# Diagnostics of COVID-19 Based on CRISPR–Cas Coupled to Isothermal Amplification: A Comparative Analysis and Update

**DOI:** 10.3390/diagnostics12061434

**Published:** 2022-06-10

**Authors:** Armando Hernandez-Garcia, Melissa D. Morales-Moreno, Erick G. Valdés-Galindo, Eric P. Jimenez-Nieto, Andrea Quezada

**Affiliations:** Laboratory of Biomolecular Engineering and Bionanotechnology, Department of Chemistry of Biomacromolecules, Institute of Chemistry, National Autonomous University of Mexico, Circuito Exterior, Ciudad Universitaria, Coyoacan, Ciudad de Mexico C.P. 04510, Mexico; mel.morales0797@gmail.com (M.D.M.-M.); erickgvaldes@gmail.com (E.G.V.-G.); paveljn14@gmail.com (E.P.J.-N.); andreagtzq@gmail.com (A.Q.)

**Keywords:** CRISPR–Cas, SARS-CoV-2, molecular diagnostics, isothermal amplification, comparative analysis, nucleic acid detection

## Abstract

The emergence of the COVID-19 pandemic prompted fast development of novel diagnostic methods of the etiologic virus SARS-CoV-2. Methods based on CRISPR–Cas systems have been particularly promising because they can achieve a similar sensitivity and specificity to the benchmark RT-qPCR, especially when coupled to an isothermal pre-amplification step. Furthermore, they have also solved inherent limitations of RT-qPCR that impede its decentralized use and deployment in the field, such as the need for expensive equipment, high cost per reaction, and delivery of results in hours, among others. In this review, we evaluate publicly available methods to detect SARS-CoV-2 that are based on CRISPR–Cas and isothermal amplification. We critically analyze the steps required to obtain a successful result from clinical samples and pinpoint key experimental conditions and parameters that could be optimized or modified to improve clinical and analytical outputs. The COVID outbreak has propelled intensive research in a short time, which is paving the way to develop effective and very promising CRISPR–Cas systems for the precise detection of SARS-CoV-2. This review could also serve as an introductory guide to new labs delving into this technology.

## 1. Introduction

Coronaviruses have caused important outbreaks in recent years, for example, Severe Acute Respiratory Syndrome-related coronavirus (SARS–CoV) in 2002, Middle East Respiratory Syndrome-related coronavirus (MERS–CoV) in 2010, and, most recently, Severe Acute Respiratory Syndrome coronavirus 2 (SARS-CoV-2), the etiologic agent of the Corona Virus Disease 2019 (COVID-19). COVID-19 was first reported in December 2019 by the Chinese Center for Disease Control and Prevention attending Wuhan local health facilities [1]. By the time of writing this review, the World Health Organization has confirmed more than 508 million cases and 6.2 million deaths [2]. The betacoronavirus SARS-CoV-2 virions (60–140 nm in diameter) are composed of a single-stranded positive-sense RNA molecule packed in a coating protein and enveloped into lipids (Figure 1A) [3]. The virions characteristically display on their surface pendant “spike” proteins which play a key role in the binding and entry to the host human cells [4]. The ~29.9 kb SARS-CoV-2 genome encodes for 13–15 Open Reading Frames (ORFs) that express a total of 12 proteins, including the non-structural ORF1a and ORF1b and the structural envelope (E), membrane (M), nucleoprotein (N), and spike (S) proteins (Figure 1A) [5,6].

The fast propagation and global distribution of COVID-19 have spurred intensive research that aims to develop novel diagnostic methods that could assist in detecting new variants and stopping their propagation [7]. Diagnostic methods based on Clustered Regularly Interspaced Short Palindromic Repeats and CRISPR–associated proteins (CRISPR–Cas systems) are among the most promising (Figure 1B) [8,9,10,11,12,13]. This is because they can potentially comply with the “*ASSSURED*” features: *Accurate*, *Specific*, *Sensitive*, *Simple*, *Rapid*, *Equipment-free*, and *Deliverable* to end-users [14]. In addition, CRISPR–Cas systems offer low-cost reactions, are highly versatile and flexible to adapt to new virus variants [7,15,16,17] and future emerging pandemics, and are suitable for large-scale production. A remarkable feature is that CRISPR–Cas systems can be synergically combined with isothermal RNA amplification methods to bypass the disadvantages of using both techniques separately [18,19,20]. When the viral RNA is retrotranscribed into DNA and amplified using an isothermal method (Figure 1C), the CRISPR–Cas system can detect it with a specificity and sensitivity similar [21,22,23,24,25] to the benchmark or gold standard: quantitative reverse transcription Polymerase Chain Reaction (RT-qPCR) [26,27]. CRISPR–Cas methods can also be designed in a portable format without the need for sophisticated instruments and highly skilled personnel, therefore being particularly suitable for deployment in low-resource point-of-care locations (POC). Its great potential is reflected in the steady growth in the number of reports using CRISPR–Cas to diagnose COVID-19 (2, 102, and 203 references for 2019, 2020, and 2021, respectively, according to PubMed).

**Figure 1 diagnostics-12-01434-f001:**
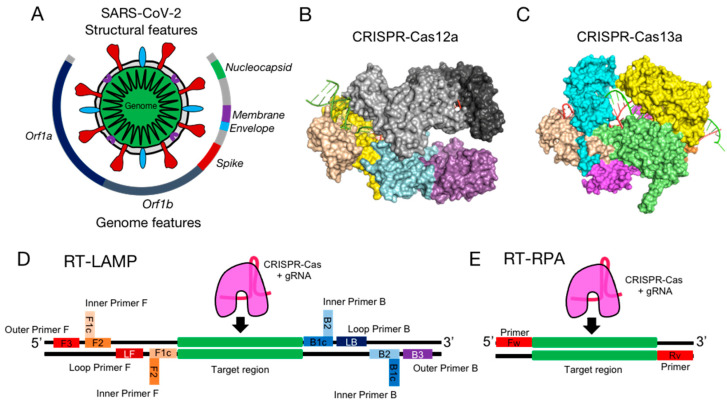
(**A**) Structural and genome features of SARS-CoV-2 virion. Class 2 CRISPR–Cas proteins extensively used in genetic diagnostics: (**B**) LbCas12a (type V, PDB: 5XUS) [28] (left) and (**C**) LbuCas13a (type VI, PDB: 5XWP) [29] (right). Colors represent different domains of Cas proteins. LbCas12a: Wedge I, II, and III (yellow), REC1 (light gray), REC2 (dark gray), PI (wheat), RuvC-I, II, and II (cyan), BH (green lime), and Nuc (magenta). LbuCas13a: NTD (cyan), Helical-1 (wheat), HEPN1-I and II (green lime), Helical-2 (yellow), Linker (orange), and HEPN2 (magenta). Schematics depicting DNA in black and the primers used for isothermal amplification methods in colors: RT-LAMP (**D**) and RT-RPA (**E**) and the target sequence for CRISPR–Cas systems (green).

In this review, we comprehensively analyzed and compared more than 50 publicly available reports that correspond to 42 different methods for detection of SARS-CoV-2 that use CRISPR–Cas systems together with isothermal amplification (in this review referred as the “CRISPR–Cas method”). From each of the 42 methods, we extract important technical details (Appendix A). All the compiled information is displayed, sorted, and ranked in Appendix A. After a systematic comparison between the methods, we found the most successful methods in terms of the time to deliver a result (time consumed in the isothermal amplification and CRISPR-based detection), the limit of detection, and the clinical sensitivity and specificity. We also discuss key methodological parameters, such as method of RNA extraction, type of isothermal method, and the CRISPR–Cas system, among others that are related to achieving a successful result. Recognizing key steps and optimizable parameters could help the CRISPR diagnostics (CRISPR-Dx) community to generate optimization and innovations that can contribute to developing more robust methods.

## 2. CRISPR–Cas in Diagnostics

The CRISPR–Cas systems are memory-like defense mechanisms present in bacteria and archaea that prevent the invasion of foreign nucleic acids such as bacteriophages or plasmids [30]. CRISPR–Cas systems recognize and hydrolyze the foreign nucleic acid through a ribonucleoprotein complex (RNP) composed of Cas proteins with endonucleolytic activity guided by a CRISPR ribonucleic acid (crRNA) [12,13,18,19,30,31,32]. The class 2 CRISPR–Cas systems, such as CRISPR–Cas9, Cas12, Cas13, and Cas14, are by far the most used in diagnostics because they only need a single Cas protein for the recognition and cleavage of the target nucleic acid sequence (Figure 1B) [11]. 

During the detection, the RNP first scans the DNA/RNA sequence to find a protospacer adjacent motif (PAM) (or a protospacer flanking site, PFS, in the RNA) and then opens the dsDNA to hybridize with the crRNA sequence to form the so-called R-loop [33]. Cas12 (type V), Cas13 (type VI), and Cas14 (Type V) differentiate from Cas9 (type II) because after cutting the specific target sequence (cis cleavage), they also cut proximate single-stranded DNA (ssDNA) (Cas12a and Cas14) or ssRNA (Cas13a) through a trans-cleavage activity [12,13,18,19,32]. This very effective collateral activity is what most CRISPR-based diagnostic systems exploit to generate a fluorescent or detectable signal through the degradation of a nucleic acid probe bi-labelled with a fluorophore and a quencher. This probe can also be adapted to be used in lateral flow strips, which allow direct visual detection.

## 3. Nucleic Acid Isothermal Amplification in Diagnostics

The pre-amplification of the target nucleic acid by isothermal methods can increase the analytical sensitivity of detection of CRISPR–Cas systems up to 10^9^ times in one hour or less [18,32]. Besides this, isothermal methods are the preferable alternative over Polymerase Chain Reaction (PCR) in the field of diagnostics in low-resource areas because they do not require expensive laboratory equipment such as thermocyclers [34]. Since all the reactions occur at a constant temperature, the isothermal amplification of DNA can be performed in common incubators and dry block heaters or even using low-cost hand warmers [35]. Likewise, it does not need specialized laboratories with specially trained personnel, and it can deliver results in less than one hour, which greatly increases accessibility and allows it to process a higher number of samples. All these advantages have led to the application of isothermal amplification methods to detect multiple pathogens, including SARS-CoV-2 [36,37,38,39,40,41,42].

In isothermal amplification, instead of using denaturing heating cycles as in the case of PCR, DNA unwinding is achieved by enzymes with strand displacement activity that work at a constant temperature. The enzymes also initiate the amplification by enabling the binding of the primers. RNA detection can be achieved by adding a reverse transcriptase. Once the DNA amplicons accumulate in large amounts in the solution, detection is accomplished by agarose gel electrophoresis, turbidity, colorimetry, or fluorescence [43]. Detection by turbidity happens due to the accumulation of magnesium pyrophosphate (Mg_2_P_2_O_7_). If intercalating dyes, such as SYBR Green I or EvaGreen, or metal ion indicators, such as calcein/Mn^2+^ and hydroxynapthol blue dye, are used, then fluorescence or color are detected, respectively.

Some of the isothermal methods most frequently used are Recombinase Polymerase Amplification (RPA) [44] and Loop-mediated Isothermal Amplification (LAMP) [34] (Figure 1C). RPA uses a polymerase with strand displacement activity at a constant temperature between 37 and 42 °C, while LAMP is carried out at a temperature between 60 and 65 °C. RPA amplifies the target sequence by using forward and backward primers, stabilized in a complex formed by a recombinase, and a single-stranded DNA binding protein which stabilizes the interaction and allows the action of the DNA polymerase. Instead, LAMP needs *Bst* DNA polymerase and four or six specific primers grouped in pairs.

Isothermal amplification methods present the major disadvantages of using non-specific detection methods (turbidity, fluorescence, or colorimetry), which could lead to an increase in detecting false positives in case of the presence of non-specific amplicons due to cross-contamination during pre- or post-processing [43,45]. Therefore, successful isothermal methods need to carefully optimize primer design and concentration, and temperature of the method.

When isothermal amplification methods are coupled to CRISPR–Cas systems, these limitations can be overcome, besides allowing an increase in sensitivity and specificity [46]. In this format, isothermal methods exponentially pre-amplify the targeted sequence, while the detection now is left to a highly specific CRISPR–Cas system. Once the amplicon has been produced, instead of linking its detection to non-specific DNA detection, it is delegated to the programmable CRISPR–Cas system, which detects it more accurately and precisely and generates an exponential signal upon the amplified DNA. Besides, CRISPR–Cas double-checks the presence of the target sequence (firstly performed by the primers of the isothermal method). This combination has led to the most extended method format for the diagnostic of SARS-CoV-2 based on CRISPR–Cas coupled to an isothermal amplification with a reverse transcriptase (RT-RPA or RT-LAMP). For simplification, here we will refer as “CRISPR–Cas based method” or “CRISPR-diagnostics” to the combination of a CRISPR system with an isothermal method. Here, we are not reviewing isothermal-only [40,41,47,48], PCR/CRISPR–Cas [49] or amplification-free CRISPR–Cas [50,51,52] methods to detect the SARS-CoV-2.

## 4. General Procedure to Detect SARS-CoV-2 with CRISPR–Cas

The whole workflow of the CRISPR-based methods comprises five general steps: (1) collection of the clinical sample, (2) preparation of the viral genomic RNA, (3) isothermal amplification of the targeted sequence, (4) target recognition and generation of a molecular signal by the CRISPR–Cas system, and (5) signal read-out (Figure 2).

### 4.1. Step 1: Collection of Clinical Sample

Sample collection is of the utmost importance to avoid misleading results since it aims to pick up the virus while preserving the integrity of the genomic RNA [53,54]. Furthermore, it needs to be managed without putting at risk the healthcare provider. Since the SARS-CoV-2 infects cells from the upper respiratory tract, most of the samples are taken through nasopharyngeal (NP) and/or oropharyngeal (OP) swabs [55] (Appendix A). The swabs are commonly stored and transported in Universal or Viral Transport Medium and sent to the analysis lab.

### 4.2. Step 2: RNA Preparation

Once the sample has been collected and transported to the analysis location, it needs to be processed in order to make the viral RNA suitable for amplification and detection. The sample can be processed in two ways: (i) extraction to obtain a highly pure RNA or (ii) using a combination of chemical and physical treatments to remove viral components and release an RNA in a less pure form [38,56,57,58,59,60,61,62,63,64] (Appendix A). Extracting methods are typically based on columns and magnetic beads commercialized as kits (e.g., from Qiagen) and yield highly pure RNA [65,66,67]. They efficiently remove enzyme inhibitors and other contaminants, thus facilitating further downstream steps of the process workflow (RNA amplification and detection and signal generation). On the other hand, “releasing” methods combine chemical (e.g., lysis buffers) with physical treatments (e.g., temperature and centrifugation) to remove viral and patient-derived cell inhibitors, contaminants, and human RNAses [8,38,56,59,60,62,66,68,69]. This makes the genomic RNA available for further enzymatic steps while providing suitable conditions to components working in further steps (e.g., polymerase, Cas protein, and gRNA).

### 4.3. Step 3: Isothermal Amplification of Target Sequence

In this step, the viral RNA is retro-transcribed into DNA by a reverse transcriptase (RT), and then the target sequence is exponentially amplified by the chosen isothermal method [34,37,39,42,44,70,71,72,73,74]. This step is critical to achieving a high analytical sensitivity since it quickly makes millions of new copies of the targeted sequence. However, it might also copy incorrect sequences, thus it needs to be properly optimized. The most common isothermal methods are Reverse transcriptase Recombinase Polymerase Amplification (RT-RPA) [44,70,74] and reverse transcriptase Loop-mediated Isothermal Amplification (RT-LAMP) [34,37,39,73] (Appendix A).

### 4.4. Step 4: Target Detection and Signal Generation by CRISPR–Cas

Here, the RNP complex (Cas protein:crRNA) binds to the target sequence (previously amplified) and cleaves it through its *cis* nuclease activity [30,33]. The RNP complex has to be formed previously, usually either in parallel to the isothermal amplification step or beforehand by incubation 10–30 min at 37 °C or even at room temperature. Most of the methods based their detection on CRISPR–Cas12 or Cas13 because they present a *trans* nuclease activity that hydrolyzes the nucleic acid probe, unleashing the reporter signal to be detected (Appendix A) [18,19]. The probe is a short single-stranded oligonucleotide (ssDNA for Cas12 and ssRNA for Cas13) conjugated to a pair of reporter molecules such as a fluorophore/quencher or antigen/biotin (Appendix A).

### 4.5. Step 5: Signal Read-Out

This final step provides the diagnostic result based on fluorescence or a colorimetric band detection on lateral flow dipsticks (LF) [8] (Appendix A). In the first case, probe with a fluorophore and a quencher is used, and the fluorescence is released upon its hydrolysis by CRISPR–Cas [18,19,68]. Fluorescence can be detected by plate readers, real-time thermocyclers, cuvette-based fluorimetry (especially in kinetic studies), or by portable UV/blue transilluminators for naked-eye detection or coupled to cell phone detection. LF requires a probe conjugated to an antigen (usually fluorescein -FAM-) and biotin. Incubating the dipstick in a solution with the hydrolyzed probe, a test red-like color band detectable by the naked-eye appears. LF are simple to incorporate in portable methods deployable in POC.

Most of the methods will deliver a qualitative result (“presence” or “no presence” of the targeted viral RNA). However, there are very few CRISPR–Cas methods that allow quantification of the SARS-CoV-2 molecules [75], although they are complex and difficult to scale-up.

## 5. Key Experimental Parameters

Within each of the five steps that depict the whole detection process, there is an inherent methodological complexity (Figure 2). We found several key components and conditions within each step are related to the observed performance and final result and show significant variability across the analyzed methods (Table 1). Recognizing the importance of these parameters could guide further optimization, leading to improving the experimental outputs of the detection methods such as total time, the limit of detection, sensitivity, and specificity.

### 5.1. Type of Sample

More than 80% of the reports have validated their methods using some type of clinical sample (Appendix A). The most common types are nasopharyngeal (NP) and/or oropharyngeal (OP) swabs. Likewise, saliva and sputum samples (15% of the analyzed methods) [59,62,75,76,77,78], which allow self-sampling directly by the patients [79,80,81], have been used successfully to detect the virus. Other less-used types of samples include bronchoalveolar lavage fluid, anal swabs, stool [76,77], and harvested lentivirus samples [82].

### 5.2. Method of Preparation: Extraction vs. Release of DNA/RNA

The viral RNA “extraction” methods, typically based on kits of columns and magnetic beads, warrant an RNA of high purity. However, using commercial kits is costly, and access to them could be limited when their global demand is high. On the other hand, “release” methods, which combine chemical (e.g., lysis buffers) with physical treatments (e.g., heat inactivation), are simpler but could potentially carry inhibitors and put at risk the optimal progress of the detection method. We found that >70% of all reviewed reports used extraction methods. This probably happens because they are commercially accessible and some of them have been approved for SARS-CoV-2 detection by local or international regulatory organizations. “Release” methods, instead, have been explored in ~30% of the reports [28,38,59,60,61,62,63,64,75,82,83]. Mostly because they are cheaper than commercial kits. They can be homemade from commercially abundant compounds such as TCEP, DTT, EDTA, Triton X-100, and Proteinase K, helping to reduce dependency on costly commercial extraction kits. Furthermore, they present other advantages such as requiring simple heat-inactivation protocols which usually incubate the sample at different temperatures between 40 and 95 °C for some minutes (5–20 min). Improving “release” methods is a very important step ahead for CRISPR-Dx; however, they need to be carefully validated in the lab and in clinically relevant conditions to confirm their reliability and reproducibility.

Besides using RNA extracted from patients, synthetic DNA- and/or RNA-encoding viral genes are required to use during the characterization and optimization of the diagnostic method, and it is necessary to carry out systematic and controlled-conditions experiments. For example, synthetic nucleic acids alone [20,35,84] or after being spiked into cells [59], biological matrices, or fluids (e.g., human saliva or sputum) [64] are used to determine the limit of detection (LoD) of the method (analytical sensitivity). Spiked samples with RNA are also used as a surrogate of clinical samples (contrived specimens) [20,59,64].

### 5.3. Targeted Genes

CRISPR–Cas systems can essentially target across the SARS-CoV-2 genome. There is on average 1 PAM site across the reference SARS-CoV-2 genome every 22.2 and 15 bp for Cas12a (PAM: 5′-TTTV-3′) and Cas9 (5′-NGG-3′), respectively. However, the gene that has been detected the most is N in ~50% of reports. The large interest in gene N may be due to the fact that it is the gene targeted in the CDC’s Diagnostic Test for COVID-19 [85]. Other targeted genes include Orf1ab (~21%), S (~20%), E (~10%), RdRp, M, Orf3, and others (Figure 1A). Genes E and RdRp are recommended by institutions such as the World Health Organization [86]. Gene S is also very important because it helps to discriminate between variants of interest and concern [7,15,16,17] Thus far, it is not clear how mutations, secondary structures, and architecture of the SARS-CoV-2 genome and its transcriptome [6] could affect how the different regions are amplified by the isothermal methods and detected by CRISPR–Cas systems [7]. However, the FDA has reported that some RT-qPCR tests were expected to fail to detect the SARS-CoV-2 Omicron variant due to deletions in the N-gene [7]. By this token, isothermal methods amplifying similar regions could expect to fail as well.

### 5.4. Type of Isothermal Amplification Method

The most-used isothermal method is RT-RPA (~50% of reports), followed by RT-LAMP (~38%) (Appendix A). The former method is clearly the preferred choice; however, alternative methods include Recombinase Aided Amplification assay (RT-RAA), Multiple Cross Displacement Amplification (RT-MCDA), and Dual-Priming Isothermal Amplification (RT-DAMP). Besides, the widely used and available non-isothermal end-point RT-PCR has also been used successfully together with CRISPR–Cas [27,49,87,88]. Almost all the reports use commercially available isothermal methods or components, either from New England Biolabs© (RT-LAMP) or TwistDx© (RT-RPA). It is important to consider that the intellectual property behind the RPA method belongs to TwistDx Inc. [74], whereas the patent of the LAMP technology has recently expired [73]. Yet, homemade isothermal LAMP enzymes used in the method iScan achieved a significant analytical sensitivity of 10 viral copies per reaction and 100% of clinical specificity and sensitivity [89], indicating that homemade productions of isothermal enzymes perform effectively, efficiently, and could reduce costs.

### 5.5. Time and Temperature of Isothermal Method

The duration time for RT-LAMP was between 20 and 40 min with a median of 30 min (before detection with CRISPR–Cas), while for RT-RPA it was 15 to 30 min with a median of 25 min. The former was incubated between 59 and 65 °C (median of 62 °C), while the latter was between 37 and 42 °C (median of 42 °C). This means that RT-LAMP takes longer than RT-RPA. Furthermore, RT-RPA can be run in parallel (one-pot test) with CRISPR–Cas systems from mesophilic microorganisms. For RT-LAMP, the only way to circumvent the temperature limitation and to develop one-pot methods is using temperature-resistant EnAsCas12a (works at 60 °C) or thermostable AapCas12b, which run simultaneously with the RT-LAMP at 59–62 °C [76,83,89,90].

### 5.6. CRISPR–Cas Systems

Most of the used CRISPR–Cas systems present collateral activity (~90%) (Appendix A). Approximately ~74% use a CRISPR–Cas12 system, either Cas12a (65%) or AapCas12b (10%), while 16% use CRISPR–Cas13 systems. Alternative systems are Cas9 (5%), Cas3 combined with multiprotein Cas complex (Cascade) from *E. coli* [91] (CONAN method), and Cas10 [22]. 

The most preferable Cas12a protein comes from *Lachnospiraceae bacterium* (LbCas12a), followed by *Acidaminococcus sp* (AsCas12a), and just a few use Enhanced-AsCas12a or from Alicyclobacillus acidiphilus (AapCas12b). Regarding Cas13a, most methods are based on *Leptotrichia wadei* LwCas13a, followed by *Leptotrichia bucalis* LbCas13a. The predominance of LbCas12a is based on its commercial accessibility through vendors such as NEB and Integrated DNA Technologies (IDT). Cas13a is usually biosynthesized locally by the lab because it is less commercially available, although recently it has started to be commercialized as well. Furthermore, Cas12a needs a shorter crRNA (~40–44 nt) than Cas13a (~54–58 nt) or Cas9 (~100 nt). Additionally, CRISPR–Cas12 detects DNA targets directly, whereas Cas13 detects RNA targets, and thus it requires the transcription of the pre-amplified DNA into RNA using T7 RNA Polymerase. Another advantage of Cas12a is that it can successfully detect DNA with improved sensitivity using engineered hybrid crRNA–DNAs [60] or Mn^2+^ ions [78]. Interestingly, the most frequently used concentrations of Cas and gRNA are 50 and 62.5 nM, while their intervals are 2.2–1000 and 20–1000 nM, respectively. The RNP ratio (Cas:gRNA) interval is 0.07–17.5, with RNP = 1 the most frequently used ratio.

### 5.7. Type of Read-Out

Fluorescence is used as a read-out in ~88% of the methods, with lateral flow dipsticks in ~48% (Appendix A). However, ~40% used both. Other seldom-reported read-out types include fluorescence anisotropy [92] and electrophoresis in gel [84,93]. For methods that incorporate Cas9 or systems lacking collateral activity, clever detection methods such as ELISA-like or conjugation of the gRNA and dCas9 with chemical groups detected by lateral flow strips have been applied [23,93]. 

Fluorescence can be detected by dedicated specialized equipment or by low-tech solutions that are suitable for resource-constrained settings. In the first case, there are plate readers, real-time PCR thermocyclers, cuvette-based fluorimetry, or sophisticated high-throughput fluorescence set-ups such as Digital PCR chips [28,75] (especially for kinetic studies and fluorescence quantification). In the second group, we can find UV/LED blue light transilluminators used in portable methods that allow direct detection with the naked-eye [35,38,77,78,94,95] or with cell phones [62,96].

On the other hand, colorimetric LF assays have been widely explored in diagnostic methods aiming to be fully deployable in points-of-care [8,83,97]. LF assays are also suitable for deployment in resource-constrained settings. The most reported dipsticks are the commercial *HybriDetect* strips from Milenia Biotech© which are specially designed to work with reporter probes conjugated with FAM and biotin. The almost complete dependence on one provider of the strips is an issue that accounts for the bulk costs and also jeopardizes their availability during high-demand times. The development of simple methods to produce strips cheaply and massively, allowing flexibility to incorporate or modify the test and control lines, would be very beneficial.

### 5.8. Reporter Probe

Most of the CRISPR-Dx methods require an oligonucleotide reporter, although exceptional cases do not [23,93]. Reported probes have a length between 5 and 16 nucleotides (nt) (ssDNA for Cas12 and ssRNA for Cas13). Probes used with Cas12a are usually rich in Ts and As. For example the most commonly reported are TTATT, TTATTATT, TATTATTAT, and TTATTATTATT [20,24,28,35,89,90,92,98]. Other reported sequences are T-rich sequences such as TTTTTTT or TTTTT [82,83], or CCCCC [75]. Some have used GoTaq^®^ commercial probes [63]. Poly-U of 5 to 7 nt length have been reported for Cas13a [25,62] and CUCUCU for Cas10 [22].

Depending on the read-out, the probe is conjugated to diverse chemical groups. When signal read-out is based on fluorescent detection, the probe is conjugated to a fluorophore and a compatible quencher. The most used fluorophore is 5′ 6-FAM (λ_Abs_ = 495 nm, λ_Em_ = 520 nm), followed by HEX (λ_Abs_ = 533 nm, λ_Em_ = 559 nm) and Alex647N (λ_Abs_ = 650 nm, λ_Em_ = 665 nm). The most used quenchers are Iowa Black FQ^®^ (IDT) and Black Hole Quencher dye. For lateral flow detection with commercial strips, *HybriDetect* from Milenia Biotech©, the probe invariably contains FAM and biotin attached to the ends. The probe is used most frequently at a concentration of 500 nM (in 17% of the methods), followed by 1000 nM (12%) and 125 nM (10%). Although the probe is used at the same concentration in most reports that use both fluorescence and LF read-out, sometimes it is used at a higher concentration for LF assays.

### 5.9. Portable, Lyophilized, and One-Pot Versions

Portability is the ability of the method to execute the analysis outside a central lab. This means that a portable method allows decentralized analysis and can be carried out in a POC. Due to the interest to develop methods suitable for POC, most have developed versions that allow portability (~75%) (Appendix A). Around 47% use fluorescence (and portable transilluminators or instruments) and 53% LF assays. We have identified that ~10% of the analyzed methods used all or some lyophilized component [28,61,99,100]. CRISPR-based lyophilized kits would play a very important role in allowing full decentralization and portability since they do not require maintaining a cold chain. Instead, ~24% of the methods report being one-pot, meaning that they combine both steps of isothermal amplification and the CRISPR–Cas detection in one. This has attracted a large interest because it can avoid cross-contamination by reducing vessel opening and pipetting and also reducing the complexity and duration of the whole process.

## 6. Experimental Outputs to Compare between CRISPR-Based Methods

Delivering a correct result rapidly (either positive or negative presence of the SARS-CoV-2 virus) is the aim of any CRISPR-Dx method. Under this consideration, the critical experimental outputs to evaluate the methods are: (1) the overall time to deliver the result, (2) limit of detection (LoD), also called analytical sensitivity, and (3) clinical sensitivity and specificity.

### 6.1. Total Time to Deliver a Result

Rapid delivery of a diagnostic result makes it possible to act quickly to safeguard the life of the patient and limit the spread of the virus, in case of a positive result. From the compiled methods, we analyzed the total time taken to carry out both target amplification and detection by CRISPR–Cas (steps 3 and 4) (Appendix A). We excluded time for sample collection (step 1), RNA preparation (step 2), and read-out (step 5) since their duration time is frequently not reported, however, a typical RNA preparation with a kit for RNA extraction would add an average 20 min to the total time, although some are as short as 5 min, whereas a release method would add on average 15 min (see Appendix A).

The total time of steps 3 and 4 takes between 15 and 90 min, yet, the most frequently reported time is 45 min, followed by 40 and 60 min. When we break out by step, we find that isothermal amplification (step 3) takes from 15 to 40 min, while detection with CRISPR–Cas (step 4) takes from 2 to 30 min, with 30 min the most recurrent time in both steps. The time consumed is not the same for both isothermal methods. RT-LAMP is used from 20 to 40 min, while RT-RPA from 15 to 30 min. This indicates that step 3 can consume more time than step 4, and RT-LAMP usually takes longer than RT-RPA.

The top methods with the shortest time to carry steps 3 and 4 are listed in Table 2. We observed they have a duration total time shorter (15–36 min) than the most frequent time (45 min) when all the methods are considered. We noticed, especially at the very top, a few methods which have drastically reduced the time by combining both steps in one (one-pot methods). Likewise, the top two-step methods seem to have achieved a drastic reduction in time in any or both steps. Isothermal amplification and CRISPR–Cas detection last less than the commonly used time of 30 min. For example, some methods run step 3 for 20 min, and step 4 for 1 or 5 min. Furthermore, it seems that incorporation of microfluidic systems, heavy optimization of both amplification and detection steps, and integration of both steps 3 and 4 into a one-pot method, help to shorten the total time by up to more than 50%.

One-pot methods that use RT-RPA can be combined with mesophilic CRISPR–Cas12/13 systems because of their temperature compatibility at 37 °C. Instead, for RT-LAMP, the thermophilic AapCas12b is used, since it is stable at 65 °C. All found one-pot methods are listed in Table 3.

### 6.2. Limit of Detection (LoD)

The LoD represents the analytical sensitivity of the method, and it is a very critical parameter that indicates the lowest concentration of the target sequence that can be detected by the method. The LoD of all analyzed methods spans from 2.5 to 5000 viral copies per reaction (vc/rx) or 0.25 to 200 vc/μL and shows the wide diversity of possible CRISPR-based methods (Appendix A). In particular, it is related to variations in methodological parameters such as targeted gene, type, and duration of CRISPR–Cas and isothermal steps, concentrations of the components, the use of Mg^+2^ or Mn^+2^, incubation time of lateral flow strips to develop the color bands, and use of dual or modified gRNAs, etc.

For all analyzed methods, their average LoD is 261 vc/rx or 12.9 vc/μL (equivalent to ~21 aM) when the volume of the reaction is considered. The median LoD is 50 vc/rx (2.5 vc/μL, ~4.1 aM) and the most frequently reported values are 200, 100, and 10 vc/rx. All these values are within LoDs experimentally determined for RT-qPCR [103]. Some methods such as CALIBURN [77] that used saliva or sputum samples report a low LoD of 2.5 vc/rx (0.25 vc/μL). In general, methods that used releasing RNA procedures presented higher median LoD (5 vc/rx) than extraction-based methods (3 vc/rx). However, VaNGuard and deCOVID methods showed good results (2 and 1 vc/μL, respectively) [28,60]. When we consider the type of isothermal method used, RT-LAMP has a median LoD of 2 vc/μL, whereas for RT-RPA it is 2.5 vc/μL. One-pot methods have an average and median LoD of 80.3 and 33 vc/rx, while for two-step methods they are 317 and 100 vc/rx, respectively. Cas12-based methods have a lower median LoD (20 vc/rx or 2 vc/μL) than Cas13a (200 vc/rx or 10 vc/μL). Interestingly, top methods classified by LoD (listed in Table 4) have a LoD < 10 vc/rx, which is equivalent to <1 vc/μL when considering the total volume used in each reaction, and represents a sub-attomolar concentration (<10^−18^ M).

According to data compiled by Soroka and collaborators [36], the average LoD value for RT-LAMP alone for detection of SARS-CoV-2 is 17.5 vc/μL while for CRISPR-based methods it is 12.9 vc/μL (Appendix A). This indicates that analytical sensitivity is improved when isothermal methods and CRISPR–Cas systems are combined.

**Table 4 diagnostics-12-01434-t004:** List of CRISPR-based methods with the lowest limit of detection (LoD).

Name Method	Acronym	LoD (vc/rx)	LoD (aM)	Iso-thermal	CRISPR–Cas	Read-Out	Total Time Steps 3 and 4 (min)	One-Pot	Targeted Gene(s)	Cas Conc (nM)	gRNA Conc (nM)	RNP Ratio	Probe Conc (nM)	Reference
Rapid and Sensitive Detection of SARS-CoV-2 Using CRISPR *	NA	2	0.17	RT-RPA	LbCas12a	F/LF	60	Not	M, N, and S	640	640	1	800	[63]
CRISPR-based Diagnostic for COVID-19	CRISPR–COVID	2.5	0.17	RT-RPA	Cas13a	F	40	Not	Orf1ab and N	66.7	33.3	2.00	166	[25]
Cas12a-linked Beam Unlocking Reaction	CALIBURN	2.5	0.42	RT-RPA	LbCas12a	F	60	Not	Orf1ab, S, E, M, and N	NR	100	NR	1250	[77]
ENHanced Analysis of Nucleic acids with CrRNA Extensions	CRISPR–ENHANCE	3	0.10	RT-LAMP	LbCas12a	F/LF	35	Not	N	60	120	0.5	500	[102]
One-Pot Visual RT-LAMP–CRISPR	opvCRISPR	5	0.13	RT-LAMP	LbCas12a	F	45	Not	N and S	200	600	0.33	2000	[94]
All-In-One Dual CRISPR–Cas12a Assay	AIOD–CRISPR	5	0.33	RT-RPA	LbCas12a	F	40	Yes	N	76.8	38.4	2	400	[35]
CRISPR-powered COVID-19 Diagnosis and CRISPR-based Fluorescent Detection System	CRISPR–FDS	5	0.28	RT-RPA	LbCas12a	F	40	Not	N and Orf1a	33.3	30	1.11	667	[104]
Multiple Cross Displacement Amplification with CRISPR–Cas12a-based Detection	COVID-19 MCCD	7	0.58	RT-MCDA	LbCas12a	LF	45	Not	Orf1ab and N	75	100	0.75	10,000	[24]
CRISPR-mediated Testing in One-Pot	CRISPR–top	10	0.66	RT-LAMP	AapCas12b	F/LF	40	Yes	Orf1ab and N	16	24	0.67	2000	[76]
In vitro Specific CRISPR-based Assay for Nucleic Acids Detection	iSCAN	10	0.30	RT-LAMP	LbCas12a, AacCas12b, AapCas12b	F/LF	60	Yes	N and E	250	250	1	500	[89]
CRISPR/Cas12a-based Detection with Naked-Eye Read-Out	CRISPR/Cas12a–NER	10	0.83	RT-RAA	LbCas12a	F	45	Not	Orf1ab, N, and E	70	1000	0.07	NR	[95]
Synthetic Mismatch Integrated crRNA-Guided Cas12a Detection	symRNA–Cas12a	10	0.83	RT-RPA	LbCas12a	F	45	Not	E and S	70	1000	0.07	0.025	[105]

* We use the name of the article because the method does not have a name. NA: Not Applicable. vc/rx:viral copies per reaction. aM: Attomolar. Conc: Concentration. F: Fluorescence, LF: Lateral Flow.

### 6.3. Clinical Sensitivity and Specificity

Sensitivity tells us about the ability of the method to differentiate true positives from false positives, while specificity differentiates true negatives from false negatives. In particular, sensitivity is the probability of detecting true positives (sick individuals), and specificity is about detecting true negatives (healthy). It is important to keep in mind that the values of sensitivity and specificity are directly related to the degree of sickness of the patient, the type of RNA extraction method, and the type of isothermal method and Cas, among other things. Analyzed methods have sensitivities ranging from ~73 to 100%, with a most frequent value of 100% and a mean value of 95% (Appendix A). Specificity, on the other hand, has values that span from ~71% to 100%, with a most frequent value of 100% and a mean value of 97.5%. There is no difference in sensitivity or specificity when RT-LAMP (95.5% and 100%, respectively) or RT-RPA (96.5% and 100%, respectively) is used. Moreover, Cas12 has the same clinical specificity (100% median) and slightly higher clinical sensitivity (100%) than Cas13 (100 and 95.75%, respectively).

The data suggest that CRISPR–Cas methods are clinically reliable to detect SARS-CoV-2 because they have similar results to the results delivered by RT-qPCR (specificity and sensitivity > 95% or sometimes even 100%) (see Appendix A for specific numbers for each method). Indeed, top methods report 100% specificity and sensitivity (Table 5). However, clinical validation generally is conducted in academic laboratories on very small cohorts (ranging from 4 to 534 patients, with a most frequent value of 25 samples) (Appendix A) and with samples previously known as positive or negative. Thus, sensitivity and specificity need to be confirmed in larger cohorts in clinical settings and with blinded samples. However, some attempts to validate CRISPR–Cas methods in clinical settings using RT-qPCR tests as reference have been carried out with acceptable results (specificity and sensitivity > 95%) [21,103,106].

When compared against an isothermal method such as RT-LAMP, the sensitivity and specificity for detecting SARS-CoV-2 are 92–96% and 96–99%, respectively, according to several systematic reviews and meta-analyses [36,107,108], which do not differ from the values acquired by CRISPR-diagnostics. However, many RT-LAMP methods have used RT-PCR as a reference, which may exaggerate its performance [107].

**Table 5 diagnostics-12-01434-t005:** List of CRISPR-based methods with highest clinical specificity and sensitivity.

Name Method	Acronym	Sensitivity (%)	Specificity (%)	Clinical Samples (Number)	Positive Samples (Number)	Negative Samples (Number)	Type of Samples	Iso-Thermal	CRISPR–Cas	Read-Out	Total Time Steps 3 and 4 (min)	Reference
CRISPR-based Diagnostic for COVID-19	CRISPR–COVID	100	100	114	61	53	NP and BALF	RT-RPA	Cas13a	F	40	[25]
All-In-One Dual CRISPR–Cas12a Assay	AIOD–CRISPR	100	100	28	8	20	Clinical swabs	RT-RPA	LbCas12a	F	40	[35]
One-Pot Visual RT-LAMP–CRISPR	opvCRISPR	100	100	26	NR	NR	NP	RT-LAMP	LbCas12a	F	45	[94]
Multiple Cross Displacement Amplification with CRISPR–Cas12a-based Detection	COVID-19 MCCD	100	100	114	37	77	NP	RT-MCDA	LbCas12a	LF	45	[24]
In vitro Specific CRISPR-based Assay for Nucleic Acids Detection	iSCAN	100	100	7	5	2	NP	RT-LAMP	LbCas12a, AacCas12b, AapCas12b	F/LF	60	[89]
CRISPR/Cas12a-based Detection with Naked-Eye Read-Out	CRISPR/Cas12a–NER	100	100	31	16	15	NP	RT-RAA	LbCas12a	F	45	[95]
Digitization-Enhanced CRISPR/Cas-Assisted One-Pot Virusdetection	deCOViD	100	100	4	2	2	NP	RT-RPA	LbCas12a	F	15	[28]
Contamination-free visual detection of SARS-CoV-2 with CRISPR/Cas12a *	NA	100	100	10	7	3	NP and OP	RT-LAMP	LbCas12a	F	45	[109]
SHERLOCK	SHERLOCK	100	100	534	81	380	Surgery	RT-RPA	LwaCas13a	F/LF	55	[21]
SHERLOCK Testing in One Pot	STOPCovid	100	100	17	12	5	NP	RT-LAMP	AapCas12b	F/LF	40	[83]
Autonomous lab-on-paper platform	NA	100	100	21	8	13	Clinical swabs	RT-RPA	LbCas12a	F	40	[99]
Manganese-enhanced Cas12a	MeCas12a	100	100	24	13	11	NP and saliva	RT-RAA	LbCas12a, AsCas12a	F	45	[78]
CRISPR Optical Detection of Anisotropy	CODA	100	100	20	10	10	Clinical swabs	RT-RPA	LbCas12a	FA	20	[92]
CRISPR–Csm-based Detection of SARS-CoV-2	NA	100	100	56	46	10	NP	RT-LAMP	Cas10	F/C	30	[22]

NP: Nasopharyngeal swabs, BALF: Bronchoalveolar lavage fluid, OP: Oropharyngeal swabs, Surgery: pre-operative samples from patients undergoing surgery. Clinical swabs: Not specified. NA: Not Applicable. F: Fluorescence, FA: Fluorescence Anisotropy, C: Colorimetric, LF: Lateral Flow. * We use the name of the article because the method does not have a name. NR: Not reported.

## 7. Cost and Manufacturing

Not many details about the costs of the methods have been reported. We found just six methods that reported a cost estimation [23,35,60,64,76,90]. The costs range from 1 to 10 USD per reaction (usually only including the cost of the proteins, gRNAs, and other materials). The costs of minor equipment such as micropipettes, thermoblocks, freezers, and lyophilizers (when needed) have to be included in the total cost of implementation and in the costs involved in the regulatory evaluation. Some have reported the preliminary cost for essential apparatus such as water baths (40 USD or less) [90]. 

In general, the production of the components is scalable and suitable for mass production; however, since the CRISPR-based methods are quite novel, systematic economic and feasibility analyses must be reported. One main limitation is the elaboration of cheap lateral flow strips. Availability of cheaper options could help to reduce costs significantly, especially if paper-based microfluidics suitable for mass production is involved.

## 8. Conclusions

The onset of COVID-19 spurred intensive research to developing novel diagnostic methods based on CRISPR–Cas systems to detect the SARS-CoV-2 virus. The particular features of CRISPR–Cas systems together with the synergistic work with isothermal amplification such as RT-RPA or RT-LAMP offer many advantages which have found a niche of application with the current fast propagation and global distribution of the COVID-19 pandemic. Scientists have used CRISPR–Cas because it can quickly (in a few days) be adapted with reliability [20].

Novel diagnostic methods of SARS-CoV-2 based on CRISPR–Cas and isothermal amplification stand out over other methods because they can clearly compete with the gold-standard RT-qPCR in some features and surpass it in others. In terms of analytical sensitivity, clinical specificity, and sensitivity, the most successful CRISPR–Cas methods can detect down to 2.5 viral copies per reaction (sub-attomolar concentration) in clinical samples with 100% specificity and sensitivity. Furthermore, the methods can deliver a result as soon as in 30–40 min (when considering the RNA preparation step) using a variety of samples such as naso–, oro–pharyngeal, and anal swabs, as well as sputum, stool, and others, in one-pot and decentralized formats. Portable kits based on fluorescence or lateral flow devices have also achieved consistent high standards when critical experimental outputs such as total time, LoD, specificity, and sensitivity are compared (Table 6 shows the top-ranked methods that score the highest globally). All this clearly positions CRISPR–Cas and isothermal amplification as an alternative to RT-qPCR, particularly for decentralized detections with shorter waiting times.

Despite the rapid advances, detection of SARS-CoV-2 with CRISPR–Cas still requires much work before mass clinical application. It needs further optimization to develop robust methods and be tested against blinded samples where it is not known a priori if they are positive or negative. Monitoring experimental outputs such as total time, limit of detection, sensitivity, and specificity would be very helpful for guidance. However, more research needs to be conducted to find which experimental parameters are key and how they affect and correlate with experimental outputs. On the other hand, there is plenty of room for innovation. For example, we can foresee the engineering of the predominant Cas12a and Cas13 and the addition of other CRISPR–Cas systems that also present collateral activity, but also the adaptation of Cas systems not based on collateral activity. Furthermore, it would be beneficial to develop multiplex methods to detect several SARS-CoV-2 genes/mutations (related to variants) [15,16] and other viruses that cause COVID-19-like symptoms (e.g., influenza) in parallel. Furthermore, the incorporation of new formats that include microfluidic systems will help miniaturizing the methods and making them more accessible, cheaper, and more efficient. An important challenge is to develop quantitative methods that continue to be simple and portable. All this will continue making CRISPR–Cas methods very attractive in the current COVID-19 pandemic, and importantly, the lessons learned here will offer advantages for coming pandemics.

## Figures and Tables

**Figure 2 diagnostics-12-01434-f002:**
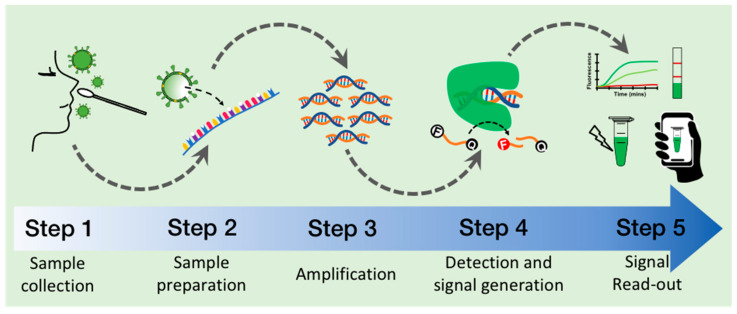
General workflow to detect SARS-CoV-2 with CRISPR-based test includes five general steps: (1) Clinical sample collection, (2) RNA preparation by extraction or release methods, (3) target sequence amplification, (4) target recognition and generation of molecular signal, and (5) signal read-out using fluorescence or lateral flow strips which could include cell phone detection.

**Table 1 diagnostics-12-01434-t001:** Key parameters and their variability across the CRISPR-based methods to detect SARS-CoV-2.

	Key Parameter (Condition/Component)	Options
Step 1	Type of sample	Nasopharyngeal and Oropharyngeal swabs, viral RNA, saliva, sputum, other
Step 2	Type and time of method of preparation	RNA extraction (5–40 min) and RNA release (5–30 min)
Step 3	Targeted Genes	Genes N, Orf1ab, S, E, other
	Type of Isothermal Amplification Method	RT-RPA, RT-LAMP, other
	Temperature	59–65 °C (RT-LAMP) and 37–42 °C (RT-RPA)
	Time	20 to 40 min (RT-LAMP) and 15 to 30 min (RT-RPA)
Step 4	Type CRISPR–Cas system	Cas12a, Cas13a, Cas12b, Cas9, Cas10, Cas3, other
	Cas protein concentration	2.2–1000 nM
	gRNA concentration	20–1000 nM
	RNP ratio (Cas/gRNA)	0.07–17.5
	Temperature	25–70 °C
	Time	1–90 min
Step 5	Type of Read-Out	Fluorescence, Lateral Flow, Fluorescence Anisotropy, Electrophoresis in gel, other
	Probe Type of Sequence Length Type of fluorophore Type of quencher	Thymine rich, Adenine/Thymine rich, Uracil-rich, other 5–16 nt FAM, HEX, and Alexa Iowa Black and Black Hole
	Format	Portable vs. Lab-based Lyophilized vs Solution-based
		Two-step vs. One-pot

**Table 2 diagnostics-12-01434-t002:** List of CRISPR-based methods with shortest total time for steps 3 and 4.

Name Method	Acronym	Total Time Steps 3 and 4 (min)	Isothermal	Isothermal Time (min)	CRISPR–Cas	CRISPR–Cas Time (min)	Read-Out	One-Pot *	Reference
Digitization-Enhanced CRISPR/Cas-Assisted One-Pot Virus detection	deCOViD	15	RT-RPA	*	LbCas12a	*	F	Yes	[28]
CRISPR Optical Detection of Anisotropy	CODA	20	RT-RPA	*	LbCas12a	*	FA	Yes	[92]
Isotachophoresis-mediated CRISPR–Cas12 DNA Detection	ITP–CRISPR assay	25	RT-LAMP	20	LbCas12a	5	F	Not	[98]
Variant Nucleotide Guard	VaNGuard	27	RT-LAMP	22	LbCas12a/AsCas12a/enAsCas12a	5	F/LF	Not	[60]
CRISPR–Csm-based Detection of SARS-CoV-2	NA	30	RT-LAMP	29	Cas10	1	F/C	Not	[22]
SHERLOCK and HUDSON Integration to Navigate Epidemics	SHINE	30	RT-RPA	*	LwaCas13a	*	F/LF	Yes	[62]
DNA Endonuclease Targeted CRISPR Trans Reporter	DETECTR **	30	RT-LAMP	20	LbCas12a	10	F/LF	Not	[101]
DNA Endonuclease Targeted CRISPR Trans Reporter	DETECTR **	35	RT-LAMP	20	LbCas12a	15	F/LF	Not	[97]
ENHanced Analysis of Nucleic acids with CrRNA Extensions	CRISPR–ENHANCE	35	RT-LAMP	20	LbCas12a	15	F/LF	Not	[102]
VirD2–dCas9 Guided and LFA-coupled Nucleic Acid Test	VIGILANT	36	RT-RPA	25	SpCas9	11	LF	Not	[23]

F: Fluorescence, FA: Fluorescence Anisotropy, C: Colorimetric, LF: Lateral Flow. * One-Pot methods combine in one single step the isothermal amplification and detection with CRISPR–Cas, hence the time for single steps cannot be reported. ** Although the methods are the same in terms of name, the methodologies are different reported by different research labs.

**Table 3 diagnostics-12-01434-t003:** List of one-pot CRISPR-based methods.

Name Method	Acronym	Isothermal	CRISPR–Cas	Read-Out	Total Time Steps 3 and 4 (min)	Portable *	Reference
Digitization-Enhanced CRISPR/Cas-Assisted One-Pot Virus detection	deCOViD	RT-RPA	LbCas12a	F	15	Yes	[28]
CRISPR Optical Detection of Anisotropy	CODA	RT-RPA	LbCas12a	FA	20	Yes	[92]
SHERLOCK and HUDSON Integration to Navigate Epidemics	SHINE	RT-RPA	LwaCas13a	F/LF	30	Not	[62]
All-In-One Dual CRISPR–Cas12a Assay	AIOD–CRISPR	RT-RPA	LbCas12a	F	40	Yes	[35]
SHERLOCK Testing in One Pot	STOPCovid	RT-LAMP	AapCas12b	F/LF	40	Yes	[83]
CRISPR-mediated Testing in One Pot	CRISPR–top	RT-LAMP	AapCas12b	F/LF	40	Yes	[76]
SHERLOCK Testing in One Pot	STOPCovid.v2	RT-LAMP	AapCas12b	F/LF	45	Yes	[90]
In vitro Specific CRISPR-based Assay for Nucleic Acids Detection	iSCAN	RT-LAMP	LbCas12a, AacCas12b, AapCas12b	F/LF	60	Yes	[89]
Digital Warm-Start CRISPR Assay	dWS–CRISPR	RT-DAMP	AsCas12a	F	90	Not	[75]

F: Fluorescence, FA: Fluorescence Anisotropy, C: Colorimetric, LF: Lateral Flow. * Portable methods demonstrated the use of the method with low-complexity equipment able to work in decentralized settings.

**Table 6 diagnostics-12-01434-t006:** Top CRISPR–Cas based methods to detect SARS-CoV-2.

Method	Acronym	STEP 1	STEP 2	STEP 3			STEP 4			STEP 5			Experimental Outputs					
		Sample	Sample Preparation	Isothermal	Temp (°C)	Time (min)	CRISPR–Cas	Temp (°C)	Time (min)	Read-Out	One-Pot	Portable?	Total Time (Steps 3 and 4) (min) **	LoD (c/r)	LoD (aM)	Specificity (%)	Sensitivity (%)	Reference
Digitization-Enhanced CRISPR/Cas-Assisted One-Pot Virus detection	deCOViD	NP	R	RT-RPA	42	15	LbCas12a	RT	15	F	Yes	Portable	15	15	10	100	100	[28]
CRISPR-based Diagnostic for COVID-19	CRISPR–COVID	NP and BALF	E	RT-RPA	42	30	Cas13a	42	10	F	Not	Lab	40	2.5	1	100	100	[25]
All-In-One Dual CRISPR–Cas12a Assay	AIOD–CRISPR	C	E	RT-RPA	37	40	LbCas12a	37	40	F	Yes	Portable	40	5	2	100	100	[35]
One-Pot Visual RT-LAMP-CRISPR	opvCRISPR	NP	NR	RT-LAMP	65	40	LbCas12a	37	5	F	Not	Lab	45	5	0.8	100	100	[94]
Multiple Cross Displacement Amplification with CRISPR–Cas12a-based Detection	COVID-19 MCCD	NP	NR	RT-MCDA	63	35	LbCas12a	37	5	LF	Not	Portable	45	7	3.5	100	100	[24]
CRISPR/Cas12a-based Detection with Naked-Eye Read-Out	CRISPR/Cas12a–NER	NP	E	RT-RAA	39	30	LbCas12a	37	15	F	Not	Portable	45	10	5	100	100	[95]
CRISPR Optical Detection of Anisotropy	CODA	C	E	RT-RPA	42	20	LbCas12a	42	20	FA	Yes	Portable	20	150	30	100	100	[92]
Contamination-free visual detection of SARS-CoV-2 with CRISPR/Cas12a *	NA	C	NR	RT-LAMP	65	40	LbCas12a	37	5	F	Not	Portable	45	20	4.5	100	100	[109]
In vitro Specific CRISPR-based Assay for Nucleic Acids Detection	iSCAN	NP and OP	E	RT-LAMP	62	60	LbCas12a, AacCas12b, AapCas12b	62	60	F/LF	Yes	Portable	60	10	1.8	100	100	[89]
Autonomous lab-on-paper platform *	NA	C	E	RT-RPA	37	15	LbCas12a	37	25	F	Not	Portable	40	100	40	100	100	[99]
SHERLOCK Testing in One Pot	STOPCovid	NP	R	RT-LAMP	60	40	AapCas12b	70	40	F/LF	Yes	Portable	40	100	20	100	100	[83]

NP: Nasopharyngeal swabs, BALF: Bronchoalveolar lavage fluid, OP: Oropharyngeal swabs, Surgery: pre-operative samples from patients undergoing surgery. C: Clinical swabs not specified. NA: Not Applicable. R: Release methods, E: Extraction method, NR: not reported, RT: Room temperature, F: Fluorescence, FA: Fluorescence Anisotropy, LF: Lateral Flow. * LoD: Limit of Detection. * We use the name of the article because the method does not have a name. ** Add 20 or 15 min extra considering the RNA extraction or release, respectively.

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
