# Peer review of "Diagnostics of COVID-19 Based on CRISPR–Cas Coupled to Isothermal Amplification: A Comparative Analysis and Update"

_diagnostics, 2022, doi:10.3390/diagnostics12061434_

Round 1
Reviewer 1 Report
The SARS-CoV-2 pandemic has given a massive push to the development of new and evaluation of existing diagnostic methods. A large number of new studies have appeared and these studies are, for the first time, comparable to each other as they are all addressing the same diagnostic target.
The authors have reviewed 42 original studies that combine isothermal amplification methods with CRISPR-based DNA recognition for the detection of SARS-CoV-2. Their review is well written and informative. In supplemental Table 1 the authors moreover provide a compilation of all studies together with key experimental parameters.
Major issues / improvements:
Missing tables -- The manuscript refers to several tables (Table 2 - 5) which I could not find neither in the main manuscript or elsewhere.
Foundational References -- The basic principle of Cas-based RNA/DNA detection was established with the two seminal papers by Abudayyeh et al. 2016 and East-Seletsky et al. 2016 in Science and Nature, respectively. Please add these references and give them some prominence in the review.
Discussion of RNA extraction -- A problem I have with almost all of the current CRISPR-Dx papers is that they downplay or outright actively hide the need for RNA extraction and how this affects the overall time, cost and practicality of the assay. To their credit, the authors clearly discuss RNA preparation as a necessary step. However, they do not include this step in the time-to-result discussions (section 6.1). Classic RNA extraction will add in the order of 20 min (manual) to 1 h (for automated protocols) and considerable cost and complexity to the test. The "release" protocols may be faster but centrifugation, heat inactivation etc will probably still take at least 15 min. The fact that most studies do not report on this, is not a good reason to not consider it. I would ask the authors to (1) critically discuss this issue, (2) correct times to result in their tables accordingly (see below) (3) add a time estimate for RNA extraction / release in Table 1, Step 2, and (4) please refrain from repeating claims of results arriving within minutes if purified RNA was the starting point (or at least add qualifying statements).
Fair discussion of isothermal-only detection -- Promising results of using RT-LAMP or related isothermal methods as a stand-alone detection method were published early on during the pandemic (and some protocols completely skip RNA extraction). This method was, to my knowledge, used at large scale in the UK and is the basis of several diagnostic POC devices. The generic and purely qualitative claim that RT-LAMP & Co show lower sensitivity or specificity is therefore not quite satisfying. The authors give a brief qualitative discussion of RT-LAMP-only detection in lines 120ff. I would like to see some hard numbers and (sensitivity, specificity) references on how well this method can perform on SARS-CoV-2 patient samples with and without prior RNA extraction. I understand that this is not the main topic of the review but a brief summary of the state of the art in "isothermal-only" detection is needed to put the results of isothermal + CRISPR-Dx into perspective.
Discussion of blind testing -- Related to the previous, due to the early stage of development, most CRISPR-Dx studies understandably report performance using non-blinded samples (positive and negative samples are know a-priory). Have any studies used blind tests? If so, these should be highlighted as their results are likely more realistic. If not, please mention this as a current limitation.
Summary figure -- it would be highly informative if the authors could summarize their review with a plot of LOD versus time-to-result (taking into account extraction as discussed above) for all the studies examined. Please add one or two representative RT-LAMP-only and RT-PCR studies into the plot for comparison.
Minor issues and improvements
Supplemental Table 1:
-
- Please include a column with at least first author name and year next to the title of each study
- Add an "estimated total time" column that adds up amplification and detection time plus a reasonable estimate for the chosen RNA preparation method
- 100% Sensitivity and Specificity values are rather arbitrary for sample numbers below, say, 20. The authors' ranking here penalizes studies with larger sample size that presumably give more realistic numbers than studies looking at, for example, only four clinical samples, as is the case for the top-ranking method. This has to be considered in the ranking somehow. I leave the how to the authors. Possibilities are a hard cut-off at, for example, n=20; by re-calculating specificity and sensitivity assuming sample n+1 would have given a false positive, respective false negative result or by other statistical means.
- Please annotate column headings in the "Evaluation" part with descriptions / explanations (e.g. using the commenting function or footnotes) as it is not very clear what e.g. "Total time %" means.
Section 3
It is not always clear whether discussion here refers to e.g. isothermal-only detection or isothermal + CRISPR-based detection. Moreover, references are given for detection "in one hour or less" in line 98 but then "it" (isothermal by itself?) "... can deliver results in a few minutes." is claimed without reference in line 104.
Section 4
Not a single reference is given in all of section 4. Especially sub-section 4.2 needs, minimally, references for regular RNA extraction and "release" methods. 4.3 requires some references for RT-LAMP and RT-RPA etc.
Section 5.2
Please give some example references for last paragraph.
Section 5.6 last sentence
"being 1 nM the most used" is grammatically wrong. Besides, I am pretty sure the authors mean a ratio of 1 which would not have any unit.
Section 6.3, line 458 -- please add some numbers/detail what "acceptable results" means.
8 Conclusions
line 488 -- "results as soon as 15-20 min" please consider my remarks regarding RNA extraction time
Apologies for the relatively long list of requests. This is not meant to distract from the fact that the authors have already delivered an excellent and very useful work. I am looking forward to the revised version.
Reviewer 2 Report
The review of Hernandez-Garcia gives an overview of the isothermal amplification methods for the detection of SARS-CoV-2 in patient samples, and their potential use in diagnostics. The review is interesting and timely. However, some issues should be addressed before publication.
Rapid antigen tests and qRT-PCR appear to be to methods of choice for quick self-diagnosis and subsequent clinical confirmation of SARS-CoV-2 infection, respectively. Both of these techniques should be presented in some detail and discussed. Specific applications where isothermal amplification methods may be preferable to these two methods should be identified.
Figure 1. In panel A, the colour coding for the genome features and the virion do not match, e.g. the spike gene is colored red but the spike glycoprotein on the virion is coloured blue. Ensure the colours match between genome and virion. For panel B, could the Cas protein be shown in complex with it’s nucleic acid template/target? Also, the different subunits and their associated functions should be labelled. For panel C, more details are required as the current presentation is not particularly informative. Spell out all the abbreviations in the legend, currently this is not the case. What do the dotted lines represent around some primers? What are the amplification conditions? Where does the Cas protein fit into the method?
A table or figure detailing the different read-out formats for isothermal methods should be included, divided into those that require dedicated specialised equipment to achieve a read-out, and those that can be determined by low-tech solutions and are suitable for resource constrained settings.
Table 1. The table title and contents are split over 2 pages, which make it difficult to read. Ensure table 1 is presented on just a single page. In the text, Table 1-6 and supplementary table 1 are referred to. However, in the submitted manuscript, only table 1 and supplementary table 1 are provided. Please add the missing 5 tables, ensuring each is presented over just a single page.
The effect VOC mutations (eg Delta or Omicron) on the specificity of established isothermal methods for SARS-CoV-2 amplification should be discussed in more detail.
On lines 53-54, the authors state ‘…..CRISPR-Cas system can detect it with a specificity and sensitivity similar to the golden standard RT-qPCR.’ A reference should be provided to support this statement.
On lines 57-58, the authors state ‘Its great potential is reflected in the steady growth in the number of reports using CRISPR-Cas to diagnose COVID-19’ Numbers or references should be used to support this statement.
It should read ‘gold-standard’ not ‘golden-standard’.
Reviewer 3 Report
In this study, the authors reviewed publicly available methods to detect SARS-CoV-2 using the CRISPR-Cas diagnostics (CRISPR-Dx) technology. They also critically analyzed the steps involved in CRISPR-Dx and listed several key experimental parameters that are important for optimization. This review may provide useful insight in developing and improving the CRISPR-Dx technology but it seems the manuscript is not ready to be submitted.
Major comments:
The absence of Tables 2-6 (as mentioned in lines 396, 410, 434, 454, and 492) is totally unacceptable as they contribute major parts in this manuscript, the “section 6: experimental outputs to compare between CRISPR-based methods” is failed to be reviewed.
Although the authors pinpointed several key components and conditions for the CRISPR-Dx, the authors failed to provide useful comments or suggestions on each of the listed key experimental parameters to help readers in further optimisation.
In addition, the authors should include a new subtopic to further explain the challenges and perspectives of using isothermal amplification and CRISPR-Cas technology in diagnostic purposes, such as standardisation, sequence limitation, tolerability to mismatches/inhibitors, carryover contamination, etc.
Minor comments:
- Line 59. The authors mentioned that they had analyzed and compared >50 publicly available methods for SARS-CoV-2 detection. However, the supplementary table only included 42 publications, please revise.
- Figure 1. Please provide a high-resolution image.
- Figure 1a. Please change the figure legend “SARS-CoV2” to “SARS-CoV-2”. Please label all the essential structure and accessory proteins in the SARS-CoV-2 structure and genome. It will be ideal if the color of the SARS-CoV-2 structure is tally with the genome below.
- Figure 1b. Please provide detailed information on each color-coded domains of Cas proteins in the figure legend.
- Figure 1c. FIP consists of the F1c region at the 5’ end and the F2 region at the 3’ end. It will be good if the FIP/BIP can be labeled clearly in the figure and explained in the figure legend.
- Supplementary Table 1. The authors have evaluated the published data without any explanation in the supplementary table or text. Please briefly describe how the evaluation has been conducted in the manuscript.
Reviewer 4 Report
The gold standard for detecting the RNA of the SARS CoV-2 virus is RT-PCR. The clinical material is nasopharyngeal swabs and/or oropharyngeal swabs where the virus concentration is the highest.
In this review paper, the Authors comprehensively analyzed and compared more than 50 publicly available methods for detection of SARS-CoV-2 that use CRISPR-Cas together with isothermal amplification.
My comments are following:
1. All abbreviations used in the text for the first time must be explained in parentheses (e.g. CRISPR-Cas systems. Not every reader can be a specialist in molecural diagnostics. 2. In conclusion, the Authors should briefly present whether this method is better than qPCR and whether in their opinion it is or not an alternative to qPCR. 3. The development of new diagnostic methods is primarily intended to shorten the waiting time for the result and possibly the cost of the test.4. The new (better) method can only indirectly inhibit the spread of the infection.
Round 2
Reviewer 1 Report
The authors have addressed most of my concerns or suggestions. I think this review will be very useful for anyone developing CRISPR-based diagnostic methods.
Reviewer 2 Report
The authors have addressed all of my concerns
Reviewer 3 Report
The authors have addressed the comments adequately with significant figures/data improvement.